# Feasibility of Participatory Theater Workshops to Increase Staff Awareness of and Readiness to Respond to Abuse in Health Care: A Qualitative Study of a Pilot Intervention Using Forum Play among Sri Lankan Health Care Providers

**DOI:** 10.3390/ijerph17207698

**Published:** 2020-10-21

**Authors:** Jennifer J. Infanti, Anke Zbikowski, Kumudu Wijewardene, Katarina Swahnberg

**Affiliations:** 1Department of Public Health and Nursing, Faculty of Medicine and Health Sciences, Norwegian University of Science and Technology, N-7491 Trondheim, Norway; 2Women’s Clinic, Ryhov County Hospital, SE-55185 Jönköping, Sweden; anke.zbikowski@rjl.se; 3Department of Community Medicine, Faculty of Medical Sciences, University of Sri Jayewardenepura, Gangodawila, Nugegoda 10250, Sri Lanka; kumuduwije@gmail.com; 4Department of Health and Caring Sciences, Faculty of Health and Life Sciences, Linnaeus University, SE-391 82 Kalmar, Sweden; katarina.swahnberg@lnu.se

**Keywords:** abuse in health care, obstetric violence, labor and childbirth, prevention, intervention, participatory theater, feasibility, acceptability, quality of care

## Abstract

Women globally experience mistreatment by health providers during childbirth. Researchers have identified strategies to counteract this type of abuse in health care, but few have been evaluated. We used a theater technique, Forum Play, in a brief training intervention to increase awareness of abuse in health care and promote taking action to reduce or prevent it. The intervention was implemented in four workshops with 50 participating physicians and nurses from three hospitals in Colombo, Sri Lanka. This article reports the views of 23 workshop participants who also took part in four focus group discussions on the acceptability and feasibility of the method. The participants reported that the intervention method stimulated dialogue and critical reflection and increased their awareness of the everyday nature of abuses experienced by patients. Participants appreciated the participatory format of Forum Play, which allowed them to re-enact scenarios they had experienced and rehearse realistic actions to improve patient care in these situations. Structural factors were reported as limitations to the effectiveness of the intervention, including under-developed systems for protecting patient rights and reporting health provider abuses. Nonetheless, the study indicates the acceptability and feasibility of a theater-based training intervention for reducing the mistreatment of patients by health care providers in Sri Lanka.

## 1. Introduction

Patients experience neglect, disrespect, verbal and other types of mistreatment, or abuse perpetrated by their health providers in a wide spectrum of health settings and patient–provider relationships around the world [1,2]. We refer to this global phenomenon as abuse in health care (AHC). AHC experienced by women in childbirth, particularly disrespectful behavior and physical violence, has been documented in numerous studies [2,3,4,5,6,7,8,9,10,11,12]. It is sometimes referred to as obstetric violence or disrespect and abuse (D&A) during facility-based childbirth. AHC in childbirth is linked to poor quality of care and associated with adverse maternal and neonatal outcomes, such as unnecessary surgical operations with higher rates of complications [13].

AHC is not limited to certain countries or health care settings. The global literature reveals a range of intentional and unintentional acts of emotional, verbal, physical, and sexual violence perpetrated by health providers in a vast array of contexts that may inadvertently cause patient suffering and contravene international recommendations for optimal interpartum care [2,10,14]. We have documented transgressions of the ethics of care to patients in health care settings in the global north and south [8,15,16,17,18,19], including in Sri Lanka where nearly all women (98%) give birth in health facilities [20]. In the different contexts of our work, experiences of AHC were related to diminished trust in and avoidance of the health care system. This can result in increased suffering due to delayed diagnoses and treatment.

Researchers have identified some strategies to counteract AHC in childbirth facilities, including providing a labor companion, on-the-job mentorship to staff, health facility improvements such as use of privacy curtains, workshops on respectful maternity care, and role playing in health care education to demonstrate practical techniques for dealing with ethically-complicated situations [13,21,22,23,24,25,26,27,28,29]. These studies, like ours, are primarily exploratory, proof-of-concept studies. They require more rigorous evaluation and larger-scale implementation to assess their long-term effectiveness to achieve improvements in patient care. The paucity of this research to date may in part relate to the enormity of the task. The origins of—and conditions that give rise to—AHC vary greatly [10]. Therefore, counteracting AHC also requires a multitude of strategies [11], including at the individual level to account for variations in dispositions, life histories and belief systems; transforming social and group norms; raising awareness or cognizance of the phenomenon; addressing power inequalities (for example, between patients and health providers and between different categories of staff in a health system); accounting for and addressing the influence of other axes of social identities such as gender and age; facilitating legal and other safeguards and supportive resources to propose and enact improvements to patient care. While challenging, the lack of development and robust assessment of interventions is an impediment to improving the responses of local health sectors to address the serious health system concerns presented by AHC. This study contributes to remedying the situation.

We conducted a pilot intervention study in Sri Lanka with a pre-post evaluation design and focus group discussions (FGDs). The aim of the overall study was to assesses the potential of the intervention training method to increase staff awareness of AHC and promote taking action to reduce or prevent it. The intervention is based on an improvisational and participatory theater method known as Forum Play (FP), drawn from Augusto Boal’s “Theatre of the Oppressed” [30,31]. The underlying goal of the FP intervention is to address factors known to erode an individual’s moral resources or ‘moral imagination’ over time—terms describing an individual’s ability to envision a wide range of available alternatives to resolve an ethical dilemma in any given situation [32]. Prior studies have shown that social pressure to conform or obey, monopolizing belief systems, fear of being evaluated poorly, and witnessing the unethical behavior of co-workers, amongst other factors, can degrade health provider’s moral resources over time [32,33,34].

In Sweden, we used the FP intervention with promising results in terms of strengthening participants’ abilities to recognize AHC and improving their readiness to act against AHC in real-world situations [29,35,36,37,38]. For this study, we adapted the intervention for use with health care providers involved in maternity care in Sri Lanka. Our previous article [39] demonstrates the potential effectiveness of the FP intervention in Sri Lanka to increase participant’s knowledge and recognition of AHC. This was indicated by participants more often reporting they had been personally involved in situations of AHC three-to-four months after the workshops compared to before. The current article reports the findings from FGDs with 23 physicians and nurses who participated in the intervention. This article adds descriptive detail and insight to our previous publication by exploring participants’ perceptions of and reflections about the potential effects of the FP intervention method on attitude and/or behavior change. Additionally, in this article, we report participants’ views on the feasibility and acceptability of the FP method.

## 2. Materials and Methods

Participants were recruited through written announcements at obstetrics departments and teaching hospitals in Colombo district in Sri Lanka and word-of-mouth by our Sri Lankan co-author and collaborator, as well as the hospital directors. Physicians and nurses at three public hospitals were invited to volunteer to participate in a training workshop on one of four consecutive days over the course of one week in November 2017. In Sri Lanka, multi-professional teams provide maternity care; in our study setting, physicians and registered nurses are the primary health providers during childbirth. Twenty physicians and 30 nurses participated in the workshops [39].

Before the workshops commenced, the participants were given oral and written information about the study and completed pre-training questionnaires on their views of AHC. On the first two days, the workshops with the physicians were conducted in English by a Swedish drama pedagogue who has many years of experience using the FP intervention method in a wide range of contexts, including with health care providers. The same drama pedagogue was supported by a Sinhala-speaking research assistant to conduct the two additional workshops with the nurses in both English and Sinhala. Each workshop was four hours, and the FP method was applied in the workshops so that all participants, including the research team, were engaged in collaboratively exploring situations of AHC. The workshops started with warm-up exercises that explored different body movements, such as speed-walking and hopping. Then, in the FP intervention, several of the participants improvised short scenes depicting AHC. The other participants, initially the audience, interacted with the ‘actors’ by replacing them in the scenes and improvising new ways to behave in the situation. By changing their roles, participants also changed perspectives and shared the experiences of others. In this process, the group was able to collaboratively rehearse alternative solutions to situations of AHC. In theory, the FP method holds potential for challenging structural asymmetries, addressing communication obstacles, and exploring a variety of actions and reactions. Figure 1 is an example of one of the situations of AHC identified by the participants based on personal experiences that we explored in a workshop. Our prior publication on this pilot intervention contains an additional example and further details about the methodology [39].

At the conclusion of the workshops, two of the authors conducted FGDs in the training center with a convenience sample of 23 volunteer participants who agreed to stay for one extra hour. In total, 10 physicians (seven female and three male) and 13 nurses (all female) participated in the FGDs. They were diverse in respect to age (ranging from 26–50 years old), length of work experience (with job seniority ranging from 2–27 years), and current work setting (from public health to psychiatric wards). Most were Sinhalese-speaking and ethnicity and lived and worked in urban centers, which is representative of the population in the district. All participants gave their written informed consent for inclusion in the study before they participated.

The discussions were conducted in English with the physicians, and English and Sinhala with the nurses with language interpretation provided by the same research assistant as in the workshops. We used a semi-structured interview guide in the FGDs, which included questions on the FP intervention and AHC in general. The discussions were audio-recorded, then transcripts of the English interviews were made by the first author and by the research assistant for the two Singhalese groups.

The study was conducted in accordance with the Declaration of Helsinki, and the protocol was approved by the Ethics Committee (ERC) of the Faculty of Medical Sciences, University of Sri Jayewardenepura (project ID: 55/17).

The data was analyzed in several iterative phases of description, classification, and interpretation, guided by strategies for qualitative content analysis outlined separately by Schreier [40] and Bazeley [41]. First, during the data collection in Sri Lanka, we made written summaries of the key discussion points following each FGD to debrief with the interpreter–research assistant and record our early reflections. The formal analysis process began with the first and last authors reading these summaries alongside the full written transcripts from the four FGDs to highlight key words and concepts, perspectives, and dialogues that seemed representative of many participants, and perspectives that seemed unique or divergent. At this stage, we also began to identify particularly salient or illustrative narrative vignettes and quotations in the data. We discussed our perceptions on the topics emerging from our respective initial analyses, and the first author clustered together similar topics.

Thereafter, in a second phase of analysis, the first author systematically coded the data using NVivo 12 software (QSR International, Melbourne, Australia). The coding process entailed annotating each summary and transcript based on the initial compilation of topics that had emerged directly from the data and, subsequently, coding each document again based on the research aim. The first author labeled the annotated sections with descriptive words and phrases that summarized the key messages in the sections, then grouped these into major organizing ideas or categories. At this point, the first author also interrogated the data in the categories; for example, by noting how many participants talked about a topic, who talked about it, what was not said and reflecting on why not, and the presence of multiple perspectives on each category. Finally, the categories were connected and supplemented with illustrative quotations from the participants, keeping in mind the theoretical assumptions that guided the study and the aim to assess the potential of the FP intervention to increase staff awareness of AHC and promote taking action to reduce AHC.

## 3. Results

In analyzing the FGD data, it became clear that participants appraised the FP intervention largely as stimulating, empowering, and positive. These three categories are described in the remainder of this section. Figure 2 presents a schematic overview of the categories and related views shared by participants in the FGDs.

### 3.1. Stimulating Discussion to Increase Awareness of the Existence of Abuse in Health Care

The intervention workshops effectively stimulated dialogue amongst colleagues about the phenomenon of AHC in a manner that participants described in the FGDs as positive, comfortable and open. All workshop participants volunteered stories of AHC they had personally witnessed, perpetrated and/or experienced. In the FGDs, the participating physicians and nurses described how the intervention opened their eyes to the diverse expressions and subtleties of AHC. Many explained the intervention had brought to light the commonness of AHC—an almost everyday phenomenon in their workplaces that is uncommonly acknowledged. Two physicians in the same FGD discussed this topic in dialogue, as follows:


*“It [AHC] happens in the whole system … from the attendant … up to the consultant. If I go to a government hospital as a patient or visitor … the first thing I say is ‘I’m a doctor’. Because if I don’t say that, they talk in such a harsh way, they just bark at you. During the workshop I was thinking to myself, how would a [regular] patient, or a person who is visiting a patient in the hospital, feel about this [manner of treatment]? Because sometimes we just ask at the health counter for some basic things like how to find a ward … [and] are talked to [by staff] in such a harsh way … I think it happens a lot … that patients get verbally—not physically, but verbally—abused.”*
Physician 1 (female)


*“Yeah, in the wards, when we work there a long time, we also get adjusted to and adopt the behaviors of the others, so probably unconsciously we also take the same steps without thinking about whether we are causing harm to a patient or not. We tend to take things for granted after a while … I should say there is a chance that we might miss such things [AHC] … Sometimes it might happen that we become desensitized to the patient point of view.”*
Physician 2 (male)

There was related and significant discussion in the FGDs about why AHC happens. Many of the physicians and nurses in the groups explained that patient mistreatment occurs without intent or awareness. As in the dialogue above, and the one to follow, they described how staff behave in ways expected of them based on social and professional norms and by observing and reproducing the behaviors of others:


*“Sometimes in labor rooms they are doing it [verbally abusing patients] in good faith, to deliver the baby quickly, but they are not recognizing that they are doing a harsh thing even in good faith.”*
Physician 1 (female)


*“Yes, because that has been normalized, they have been used to it, they have observed that as trainees from their superiors.”*
Physician 2 (female)

During these discussions, some participants, notably male physicians, seemed unwilling or unable to accept personal culpability for wrongdoing or mistreatment. For example, when we asked if patient mistreatment could ever be justified, these participants diverted and obfuscated the topic and spoke about AHC as the actions of others. The dialogue below between two male physicians in one of the FGDs illustrates this type of conversation:


*“We can be abusive, but the thing is that the medical staff are also under stress in some situations, so they tend to overreact. But we can’t justify that. But as human beings, we tend to lose our tempers, and some patients are very difficult to handle”.*
Physician 1


*“In fact, most of the midwives and the staff are good—kind and good, but when they are pressed for safety, they shout, and that has become a norm it seems, probably. They also don’t realize it as abuse. The patient won’t take it as abuse either.”*
Physician 2

However, other participants—both male and female, and particularly in the two FGDs with physicians—contested their colleagues’ justifications for AHC. They argued that staff are aware of the occurrence of patient mistreatment, but AHC is maintained because most staff do not know what to do nor how to address it. One female physician described this from her personal experience:


*“A barrier to acting on behalf of patients is lack of knowledge on how to act. In my case, I think that is what happened. I had the willingness to act but I thought that the only way to act was to confront them [abusive colleagues], and by confronting, I thought it might make things worse for the patient, so I used to keep silent.”*


Several participants similarly reflected that the intervention had raised their awareness of the harms of a different kind of AHC; namely, inaction or failure to intervene when witness to patient mistreatment. For example, a female physician described that she and her colleagues:


*“Don’t intend to harm—no, there’s no intention to do harm. Everything we do, in the end, is for the betterment of the woman, but … omissions are there. I think most of the cases [of patient mistreatment] are due to omissions, not due to commissions. Like we don’t have the intent to cause any harm but not doing things you are supposed to be doing is still causing harm.”*


Several participants in the FGDs noted the intervention was valuable for them as it forced them to reconsider patient perspectives on care. These participants described feeling new empathy for patients at the workshop’s conclusion. Others felt remorse for past inaction, and some described feeling overwhelmed by the enormity of the task of addressing AHC:


*“I thought, in my reflection [watching the FP scene], I should have done more in these kinds of situations. There are patients with STDs [sexually transmitted diseases] and they deliver in the ward and, there in the labor room, even in front of the patient, sometimes they [other staff] say, ‘Doctor, put on double gloves’. They say those things for our safety, but they say those things in front of the patient, so the patient feels it because they hear it and they know. And when patients are not ‘cooperating’ [in labor], they [other staff] will say, ‘Oh they [the patient] are acting like a baby here now but they’ve done those things [have sex/become pregnant]’. Some people say those things, not everybody, but I should have done more in these situations.”*
(male physician)


*“The most important thing I realized today is that even as a medical student we could have acted in a meaningful way to prevent patients being abused. I used to watch desperately how patients were being treated when I also was a medical student, but I was under the impression that since I was at a lower level I was powerless and I would not be able to make any change. But, if I had known at that time, some things I learned today, I could have prevented many unfortunate things happening.”*
(female physician)

Thus, the intervention stimulated discussion of AHC and seemed to encouraged awareness and reflective thinking about it. Increasing awareness of the existence of AHC is essential to addressing the phenomenon, but it is only one component of sustained behavior change. In a hierarchal health care system, as in Sri Lanka, it requires courage and implies taking a risk—even in role play—to challenge situations that one experiences as negative, distressing, or morally wrong. Thus, we turn now to exploring if our data provides indications of the potential of the intervention to improve participant’s feelings of readiness to act to confront or counteract AHC.

### 3.2. Empowering Feelings of Readiness to Respond to Abuse in Health Care

Several participants in the FGDs mentioned the usefulness of rehearsing or practicing alternative strategies in the FP intervention to potentially improve patient care. They described that viewing a scenario of AHC be ‘re-played’ engages memory and encourages reflection about possible options for responding differently in similar future situations. The below quote from a male physician illustrates:


*“When this [scenario] was role-played again, I was generating options of responding in such instances, and I thought there was one instance in which I could do something … Even somebody who doesn’t have that much power can go and look at her [the patient] and reassure her, even in the presence of a consultant. So, as I watched, I thought that, even without much power in a team, you can still do some interventions which will help to change the situation.”*


The workshops were facilitated by an external pedagogue trained in the FP method, but the FP scenes were based in the participants’ realities (that is, stories of AHC the participants described and selected). Additionally, the participants contributed the available options for challenging the negative situations. Many participants in the FGDs remarked on the value of these techniques for ensuring the role plays remained realistic and relevant for their contexts. Additionally, some participants, both physicians and nurses, noted experiencing shifts in their perspectives or attitudes about the accessibility of alternative courses of behavior or action in situations of AHC. Here, a nursing matron (senior nurse) describes her feelings with us after the workshop:


*“After seeing the incidents [in the FP], I will now try to speak to the people involved [in cases of patient mistreatment]—if not directly, then indirectly. I will speak. If we consider our country, women are in a lower [oppressed] situation. There is domestic violence—it is common. I don’t want violence in the labor rooms too … for them. Violence happens at home, it happens in society, it happens in the labor room at delivery too. It is inhumane … We must try at least to prevent it in the labor rooms.”*


Our findings therefore allude to increased awareness of possibilities to act or behave differently in response to AHC. However, there were notable critical views of the potential of ours or any intervention to address AHC in a transformative way in the Sri Lankan health system. The participants who shared these opinions, all physicians, discussed at length in the FGDs about the enormity of structural change required to transform the health system in a way that would prioritize and protect patient rights. As an example, the physicians mentioned needing comprehensive and nuanced incident reporting systems to communicate concerns regarding the mistreatment of patients—accountability systems which would be taught in health care education and effectively enforced. They also talked about the prohibitive challenge it is for a patient to report abuses perpetrated by their health providers due to social inequalities. They discussed how limited the focus is in medical and nursing education on ‘soft skills’, such as communication skills. One of the male physicians was particularly vocal about these topics; here, he explains:


*“Complaints about care are rarely made by patients. In that case, the patient would have to be of a very powerful background, only then would they do that [report an incident]. The rest of the time the perpetrators get away with it … My comment is that, like she says [referring to a female physician colleague], we are concentrating on more clinical stuff in our CPD [continuing professional development] activities primarily because we can find sponsorship for these other issues. Because otherwise you will not be able to run a training course. Let’s say you wanted to have a training on abuse of women on maternity wards … There’s nothing about prescribing there. There’s nothing pharmacological. And funding is not available from our government institutions for training … so some of the programs about patient care or safety do not run because of these sorts of issues.”*


Additionally, participants shared their opinions about the difficulties of responding to AHC in the presence of other staff with more authority. The female physicians in the groups emphasized the overwhelming challenge of standing up to colleagues in their workplaces, as exemplified in this quote:


*“There’s a hierarchy in our setting, there’s a hierarchy in the ward setting, and we are afraid of breaking that hierarchy even if we see that something is wrong and a patient is treated badly … We don’t have the courage to stand up. Sometimes that [mistreatment] happens because of the hierarchy. If the higher person is treating the patient badly or talking to the patient badly we are not in a state to stand against them.”*


Many participants felt that systemic change can only happen when senior male leaders initiate it, for example by modelling respectful patient care. One of the female physicians shared her experience as a young intern, illustrating this opinion:

*“I did my internship with a very senior consultant. He was about to retire and he was very calm and quiet. He was a Muslim consultant … I don’t know, maybe because of his religion, he always respected ladies. He used to always touch the mother [in labor] and he always said, ‘you ladies are doing good, very good’, like that, he used to tell. So, as a team, we also used to do something better than the other wards where there were fewer patients … We had to do all the c-sections and deliveries, so we were very busy, so we had to prepare about 10 mothers per day, and as interns, we had to take the mothers very fast to the theater. And when one theater was going on, we had to keep the other mother at the door and another one at the waiting area … so you would think there was no time to explain all things [to the mothers] like we should have done, but I feel we did give better care than on other wards”*.

A nurse participant also shared a similar perspective:

*“If we can change the attitudes of the higher people we can change things [for patients]. For example, if we see litter and pick it up, others will follow. In our case, if the VOG sir [consultant physician] does something good, we follow. If he talks to a patient badly, it is not that we do the same, but we can’t oppose him. If he becomes a role model, others will follow”*.

The more critical views expressed in the FGDs were common and held by participants with different backgrounds. As such, they challenge generalization about the positive acceptance and potential impact of the intervention. However, even the most critical participants remarked that the intervention highlighted possibilities for small changes within their capacities that could improve patient care—for example, making eye contact with a patient or asking her name. There are many illustrative quotes from the FGDs to demonstrate this type of insight about possibilities to behave differently. Below are two female physicians in dialogue on this topic:


*“From the scenarios, we learned that we don’t have to do a big thing [to make a positive change]. Sometimes a very small thing can change the whole situation, you can make the patient happy and safe. So that’s really important because we have to see, if we don’t see, we can’t think and act also. When we make a small script, we can re-think and reproduce it later, but I think we need teamwork and leadership. Leadership is important. But I also think that rethinking that small things can make a difference, that was really good.”*
Physician 1


*“Yes, when you just changed the direction of the patient on the bed [in one of the FP scenes], it made such a big difference. And you can start your clinic that way so that problems won’t occur in the first place. It’s really important … we have to think [about the patient].”*
Physician 2

Similarly, several participants mentioned the intervention helped them to identify opportunities for system-level improvements within their existing structures. For example:


*“In front of the other staff we are uncomfortable [to intervene]. We can’t directly say opposing words but, at our weekly discussions, we could talk about it, we can say there that we should treat patients as humans … Basically, in the ward meetings [now], we discuss the clinical scenarios … we don’t discuss about these kinds of things—ethical issues … The senior consultants can take the initiative within their units … to discuss these matters in their weekly education sessions.”*
(female physician)

### 3.3. Positive Indicators of Feasibility and Acceptability of the Intervention Method

We concluded the FGDs by asking participants their opinions on the FP method. Almost all commented on the effectiveness of the participatory and visual nature of the intervention wherein participants assumed different roles in the FP scenes of AHC—as actors and/or the viewing audience. The below excerpts are examples of this type of discussion:


*“We learned a new methodological technique, Forum Play … Some of the participants became actors, like to represent or create a scenario, and others became observers and they, themselves, worked among themselves, on how to improve the situation, rather than listening to a lecture. We are used to role plays but, here, it was a different technique.”*
Physician 1 (female)


*“Yeah, it was very interesting, the different techniques used. And always we were communicating with the others, so there was a lot of interaction within the group … a lot of thinking went into it, and we discussed a lot, we had to think about real situations and tell those things. I think I learned a lot of new things here.”*
Physician 2 (female)


*“The role play, it was visual, and that goes directly to our heads, it is like it is really happening. It is like ‘live’.”*
(female nurse)

Several participants, especially the nurses, mentioned other benefits of the method that made for a good learning situation, for example elements of play, fun, and silliness that helped to encourage collective understanding and counter the seriousness of the topic of patient mistreatment. Many participants commented that the physical use of bodies in the workshop encouraged a different way of learning, engaged a different component of memory, or facilitated the expression of different emotions compared to in traditional styles of training familiar to them. The following quotes from two nurses are illustrative:


*“With the warm-up activity done before the FP, which was on body language, we got awareness already that when we communicate with people, it is not only what we speak that matters, but also the way we keep our hands and legs and touching that matters. It was helpful … [that] we think together in the play, rather than thinking alone, it is better. It is enhancing with each other’s ideas.”*



*“When we participate practically, we ‘feel’ more.”*


## 4. Discussion

The widespread problem of disrespect, undignified care, and other forms of mistreatment or abuse of women in childbirth has been clearly established in recent years, yet is often overlooked, unrecognized, or passively tolerated by the health care workers who perpetrate it or are witness to it [1,9,10]. Our participants in this study described a similar situation in their workplace contexts in Sri Lanka. However, our study findings suggest that participatory theater techniques, such as FP, hold potential as an intervention method for preventing and addressing AHC in Sri Lanka. A majority of participants attributed the high level of engagement and interaction in our workshops to the effectiveness of the FP technique to open their eyes to the commonplace nature of AHC in their current health care structures. Participants reported starting to consider the implications of the ‘normalization’ of AHC in childbirth and reflecting on the attitudes and habitual actions (including inaction) they held which may perpetuate AHC (for example, conforming to group norms and diffusing personal responsibility for improving a situation despite recognizing that a patient’s care could be greatly improved). Similar to Ratcliffe et al.’s findings [22], in their evaluation of the effects of respectful maternity care workshops, participants in our FP intervention reported leaving the training with a greater capacity to empathize with the women to whom they provide care. Together, these findings suggest that even a brief training intervention like ours can promote empowered thinking by creating opportunities for participants to reflect on and actively experiment with problems and solutions related to AHC.

The mechanisms and structures that underlie AHC are complex, as are the varied conditions in which AHC occurs [42]. As such, addressing AHC in childbirth requires a host of multifaceted interventions, ranging from small structural changes to more complex behavioral change [22,23,28]. FP, and other Theatre of the Oppressed techniques, have been applied in a diverse range of health, education and social work studies to rehearse options for solving problems at individual, group and societal levels [43,44,45,46]. To change behaviors in real-life settings, it is necessary for individuals to have perceived self-efficacy or belief in their ability to change their own behavior and/or to challenge other’s behaviors [47]. Additionally, one must have the essential behavioral capabilities, or prerequisite knowledge and skills, to perform the action [48]. One of the underlying foundations of FP is to provide an opportunity to envision and rehearse alternatives that could support a transition toward change. FP, like other techniques of Theatre of the Oppressed, involves integrating the body and the mind, thus it can be used to become aware of, but also to expand, ‘habitus’—a term referring to our deeply engrained and socially-entrenched habits and dispositions [49,50]. Much work remains to be done in developing comprehensive theater-based programs to prevent and address AHC and longitudinal evaluation of their impacts. We hope the results of this pilot study will encourage others to join us in investigating the potential of such methods for improving patient care.

Participants in our study identified various institutional barriers that affect how staff provide care to women in childbirth. Policies and accountability systems to prevent AHC need to be more detailed, nuanced, integrated in clinical education, and enforced. Patient rights are underdeveloped, and patients lack effective means to report abuses by staff as well as the power privileges to safely challenge health care providers without fearing implications for their current or subsequent care. The participants in our study held different opinions and perspectives on these topics, indicating their solutions are also complex.

It is also clear from the reports of our study participants that sustained changes to ensure respectful and dignified patient care will not happen without the commitment and support of hospital leadership and modeling of senior staff. While staff have far more rights than patients in Sri Lanka, many staff are still in vulnerable positions in a very hierarchical medical system. Participants felt that reducing AHC would require speaking out against the majority, and they feared the practical repercussions of such challenges. This is similar to what Ratcliffe et al. described in Tanzania, where “health providers themselves may also be subject to disrespectful treatment on the part of the health system, which can lead to burnout and further disrespect and abuse of patients … reinforce [ing] a vicious cycle that perpetuates the pattern of abuse” [22] (p. 2). These factors may help to explain why it seemed easier for many participants to talk about ‘others’ as perpetrating AHC and as responsible for changing the situation. Perhaps particularly in hierarchical workplace contexts, and collectively oriented societies, senior leaders have significant power to dictate the standard of patient care. For these reasons, participants in our study felt that leaders (senior consultants) should set examples for respectful care and treatment for their staff on wards. We consider it a promising result of our study that the most senior doctors and the nursing matron in the FGDs reported feeling an increased sense of responsibility to ensure respectful care. Similarly, it is promising that most participants in the study, regardless of job position or seniority, were able to identify available possibilities for improving patient care within their existing structures.

Overall, participants found many inherent benefits in the FP method. They described it as relevant, practical, beneficial, valuable, and useful for understanding and addressing AHC in their workplaces. All workshop participants shared stories of AHC willingly. From this, we can imply that the workshops encouraged a ‘comradery’ or sense of collective experience or understanding. Some of the critical ingredients for the success of our workshops undoubtedly relate to our highly skilled FP facilitator/pedagogue and language interpreter, which are factors we did not discuss with the participants in the FGDs. For other research teams considering applications of FP or other techniques of Theatre of the Oppressed in their work to address AHC, the value of a highly skilled FP facilitator cannot be overstated. The facilitator is responsible for creating a safe and open environment for the sharing of difficult stories (in some cases, working across cultural and linguistic boundaries), and an environment that is accepting of individual’s different responses to the FP exercise. Skilled language interpretation is also crucial, as is planning for the extra time required to allow for simultaneous interpretation.

From this pilot study, we have identified necessary improvements to our FP intervention before testing it on a larger scale. It would be valuable to combine the FP method with other research methods and data collection techniques—for example, participant reflective writing about the workshops; in-depth interviews; recording the FP scenes and then watching the videos with participants to reflect on what was happening and what participants in different roles were thinking—the inner monologues ‘behind the scenes’. A larger study is needed, incorporating specific measures of effectiveness, acceptability, and feasibility. The study design should be able to measure direct correlations between use of techniques, such as FP, and a decrease in abusive behaviors or attitudes or improvements in terms of quality of subsequent health care provided. Possible outcome measures for such a future study could include comparisons with reported cases of patient abuses from other hospitals that did not receive the FP intervention; provider self-reports about feelings of empowerment to take action to challenge poor caregiving practices they witness from colleagues, or increased awareness of the impact of their attitudes and behaviors on patients; or patient reports on feeling safer and more satisfied with care received after the interventions.

## 5. Conclusions

Our study findings indicate that participants viewed a brief FP workshop as a potentially effective, acceptable, and feasible training strategy for reducing the mistreatment of patients by health care providers in Sri Lanka. The FP workshops provided a forum for stimulating awareness of and questioning the normalization of AHC, as well as for rehearsing alternative behaviors in such scenarios. These are promising findings for the potential of a brief training intervention to reduce patient mistreatment in obstetric care settings in Sri Lanka, and possibly in other contexts. To establish evidence of effective and practical methods for counteracting AHC, interventions like ours need to be scaled-up and evaluated in different contexts and under varying conditions.

## Figures and Tables

**Figure 1 ijerph-17-07698-f001:**
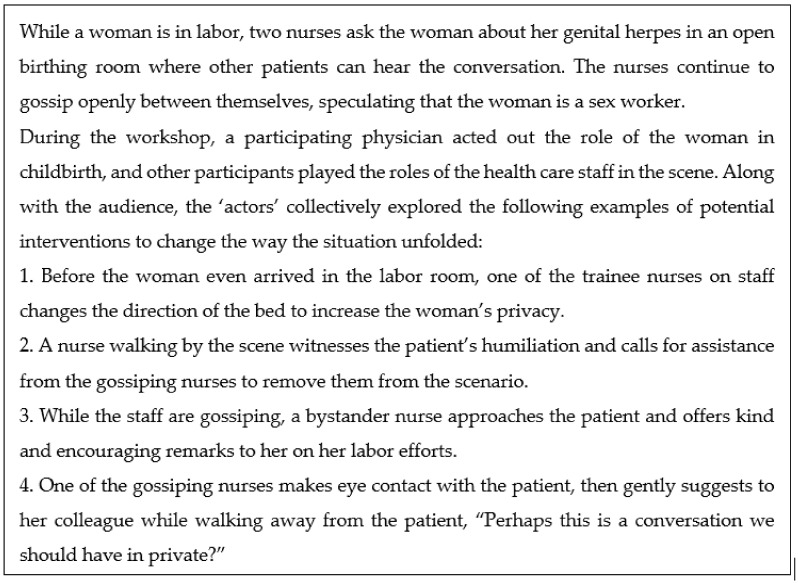
Example of a situation described by participants as illustrative of abuse in health care, and suggested solutions explored during the intervention.

**Figure 2 ijerph-17-07698-f002:**
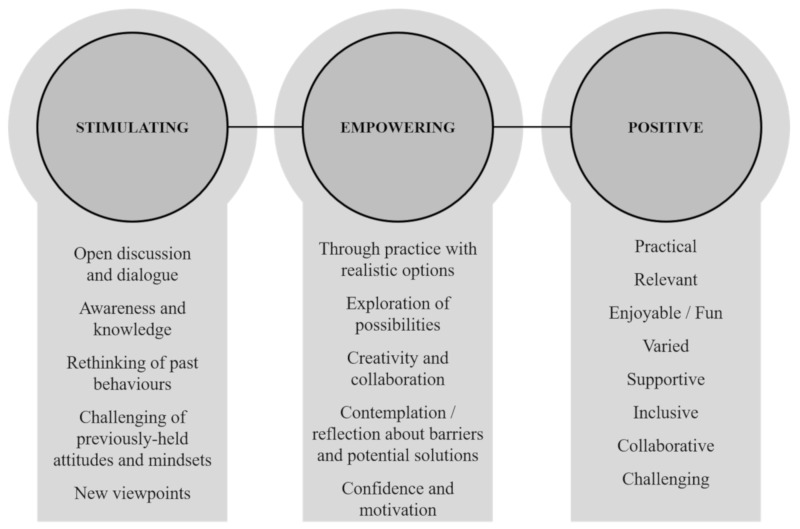
Participants’ appraisal of the Forum Play intervention.

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
