# Peer review of "Feasibility of Participatory Theater Workshops to Increase Staff Awareness of and Readiness to Respond to Abuse in Health Care: A Qualitative Study of a Pilot Intervention Using Forum Play among Sri Lankan Health Care Providers"

_ijerph, 2020, doi:10.3390/ijerph17207698_

Round 1

Reviewer 1 Report

This is a very well written paper that is a joy to read.  The study is sound and, as a pilot, offers the potential to become a landmark methodology to improve the experiences of women in labour in this difficult environment.  It is particularly gratifying to see that one of the authors is from Sri Lanka and able to ensure that the cultural suitability of both the programme content and approach to the problem are truly embedded in appropriate cultural norms. I fully support publication.

The study was well designed and executed and included a wide range of professions who were able to access the training programme, and it was particularly important that the senior staff were included in the session to provide an appropriate cultural shift within the unit and act as role models for more junior staff and students. The positive comments from the participants suggests that the content was well received and staff noted that they gained specific and useful tips on how to manage these difficult situations. As this is a pilot, I hope the team are able to find ways to scale up the training programme in order to roll it out across a wider geographical area. This is an important approach to training that has the capacity to improve the birthing experience of many mothers and I look forward to learning about how the programme is used in the future.

Author Response

Dear Reviewer,

Thank you for your encouraging review of our manuscript.

Indeed, we have various ideas for future applications of the training piloted in this research, and for other related interventions aimed to promote respectful birth care practices. That you for motivating us to take this work forward. We are grateful for your interest in the work, and for your time and efforts reviewing the manuscript.

Best wishes from the authorial team.

Reviewer 2 Report

This is a very well written paper follow up paper on the authors' previous report on their research. The research itself was carried out using the appropriate methodology. My only reservation is on the applicability of the methodology at a wider scale, and without the involvement of the researchers, but this cannot be addressed at this stage.

The paper on abuse in health care (AHC) reports on a well designed project aiming to reduce AHC especially in obstetric settings in Sri Lanka. It applies a technique already applied in Sweden and the transfer of the method to a country with a different cultural background is commendable – too often techniques in Northern and/or Western countries are recommended for wider application without testing the feasibility of the transfer. The paper is well written, with due care by the authors not to offend the participants or the process itself. The authors are aware of the difficulties, including that of translation. They followed a very detailed protocol in the design and execution of the study and in the analysis of the results. My only reservation is that the method used, Forum Play, requires special training for those who apply it, and therefore it is difficult to reproduce and apply on a wider setting. The authors are aware of the difficulty and the discussion and recommendations are carefully worded and they also make reference to other more traditional ways of dealing with such situations, such as relevant policies, accountability systems, developing patient rights and involving hospital leadership and senior staff. Finally the quality of the English language in the paper is excellent. For all the above reasons I recommended accepting the paper “as is”.

Author Response

Dear Reviewer,

Thank you for the insightful and motivating comments on our manuscript.

You raise two critical points about the Forum Play methodology that we have also considered and discussed: the scalability and the necessity of trained/skilled facilitators. We feel it is essential to involve midwives and other health providers in developing the training curriculum for local Forum Play facilitators, in order to successfully develop and deliver the training on a wider scale and to potentially extend its applications to other maternal health services. This pilot study has also led us to developing ideas for possible alternative interventions for promoting respectful birth care. Thank you again for encouraging us to take our work forward.

Best wishes from the authorial team.

Reviewer 3 Report

The work deals with a very important issue. Research was conducted in an interesting and substantiated manner. The conclusions from the research can be used in clinical practice.

I have two comments:
1. Please refer to the literature on this statement in verses 56-57
2. Do the midwives or nurses look after women in the perinatal period? If there are midwives, I believe they should be considered. If not, it is worth mentioning that nurses care for women who give birth.

Author Response

Dear reviewer,

Thank you for the positive and insightful review of our manuscript. We are motivated to take this work forward.

We have responded to your two comments in the revised manuscript.

First, regarding the references in lines 56-57, we were referring to the references cited in the previous sentence. Apologies this was not clearer. Hopefully we have clarified the text adequately now.

Secondly, maternity care is provided by multi-professional teams in Sri Lanka, and the composition of professionals varies to some extent by the location/region in the country. In some regions, public health midwives are responsible for all prenatal care, for example. In terms of perinatal care, though, it is almost exclusively provided by registered nurses, midwifery-trained nurses, and/or auxiliary nurses, particularly in our urban study setting. Therefore, we invited all categories of nurses to take part in our pilot study. We have added a sentence to this effect in the revised manuscript (lines 99-101).

Thank you again for your time and effort reviewing our manuscript.

Best wishes from the authorial team.

Reviewer 4 Report

The manuscript describes the study conducted by the authors in applying an alternative method to improve the health care quality and conditions in interpartum care. Although the approach is interesting and innovative, the study should be presented in a more rigorous scientific manner. The proofread by a native English speaker is recommended due to the typos and grammatical errors. Specific comments are reported below.

Section “Introduction”:

Page 1- line 42: What are the maternal and neonatal outcomes? How does it affect the mother and newborn health? Please, explain

Page 2 -line 46-48: does official data of prevalence of this phenomenon exist?

Page 2 – line 50: The international recommendation for interpartum care provded by the WHO should be mentioned, to provide a comparison between optimal treatment of mothers.

Page 2 – lines 55-67: please, provide references.

Section “material and method”

The data analysis should be discussed in a separate subsection

Page 3 – line 97: Please provide details of the participants, like age and sex.

Page 4 – lines 148-159: was the method applied to other studies? What is the rational applied to the data analysis? Please, provide scientific explanation and evidence of the approach used in this study.

Section results:

Number of participants belonging to specific groups should be provided to explain the efficacy of the approach, demonstrating the prevalence of specific answers. Moreover, a statistical approach to analyse the groups should be conducted.

Page 9 – lines 377 -380: Was a validated questionnaire proposed to participants to evaluate their responses? Were criteria fixed to discriminate positive answers from negative answers?

Section “discussion”:

Page 10 – line 413: Please, provide references.

Page 10 – line 416: What does “many” mean?

Page 10 – line 433: Please, follow the guideline for authors in providing in-text- citations

Page 10 – lines 433-435: Please, explain further the concept providing scientific evidences supported by references.

Author Response

Dear Reviewer,

Thank you for the interesting comments on our manuscript, and for your significant time and efforts invested in the review.

We have addressed as many of your comments as we could in the revised manuscript. We have explained below why we have not, or could not, address some of the comments. Our responses are in red text under your comments. We hope they will address your concerns satisfactorily.

Thank you for improving our manuscript.

Best wishes from the authorial team.

Section “Introduction”:

Page 1- line 42: What are the maternal and neonatal outcomes? How does it affect the mother and newborn health? Please, explain

The growing literature on abuse in health care experienced by women in childbirth points to direct and indirect adverse maternal and neonatal health outcomes, such as unnecessary episiotomies and caesarean sections, acts of physical abuse/use of physical force by health providers, abandonment during care or refusal to care (leading to fetal deaths, placental abruptions, etc.), threats/blaming/humiliation that can deter women from future use of health services, etc. We have cited 10 studies (references 2-12 in our manuscript) that document such health affects. We have now added one example in the text in the revised manuscript (refer to line 44). Additionally, we mention health affects in lines 52-53.

Page 2 -line 46-48: does official data of prevalence of this phenomenon exist?

Various studies have been carried out in recent years to document the types of and extent of mistreatment of women during childbirth. Bohren et al.’s systematic review of the literature, which we reference in the lines of the manuscript that you refer to in this comment, synthesizes the qualitative and quantitative literature (as of 2015). However, there is still no consensus at a global level on the definitions and measurement techniques for the phenomenon. As such, there is significant research today to develop typologies to effectively capture the phenomenon. This is a prerequisite to carrying out global or multi-country prevalence studies.

Page 2 – line 50: The international recommendation for interpartum care provded by the WHO should be mentioned, to provide a comparison between optimal treatment of mothers.

Thank you. We have added this now (see line 48).

Page 2 – lines 55-67: please, provide references.

We have already cited 9 studies (our references 13, and 21-29) that demonstrate the variety of interventions in development now to address AHC in childbirth contexts. We have now added two additional references (see lines 63 and 64) – the first of which is to Bohren et al.’s landmark mixed-methods systematic review which demonstrates the vast conditions in which AHC originates as well as its diverse expressions.

Section “material and method”

The data analysis should be discussed in a separate subsection

We appreciate the comment here, and we would be happy to add sub-sections to the materials and methods section at the editor’s request/recommendation (e.g. we could add sub-sections not only for data analysis, but also for recruitment, intervention technique, data collection procedures, etc.).

Page 3 – line 97: Please provide details of the participants, like age and sex.

We provide this data in lines 129-133. We could move this information earlier in the manuscript, but we feel it is adequate in its current place. As none of the other three manuscript reviewers requested that we introduce the details earlier, would you be satisfied if we defer to the editor’s guidance on this potential change?

Page 4 – lines 148-159: was the method applied to other studies? What is the rational applied to the data analysis? Please, provide scientific explanation and evidence of the approach used in this study.

Yes, the methods we used for data analysis are common techniques for qualitative data analysis, applied in many other studies, although always with some flexibility and variation. Qualitative data analysis has been likened to a craft in the sense that the same analysis methods cannot be applied, like statistical tests, in the exact way from one qualitative study to the next. This is related in part to the purpose of qualitative research, which has a discovery focus. It is also because the researcher is the ‘tool’ or ‘instrument’ of qualitative analysis, performing all the basic operations of the analysis (identifying patterns in the data, assigning significance to particular passages in the transcripts, etc.). Therefore, there is always some subjectivity and variation in the processes.

You asked specifically about the rationale for the coding process (current lines 155-166 in the revised version of the manuscript). Coding is a common approach to organise and identify themes in qualitative data. It is one way to comprehensively examine the data and start summarising the text, or grouping passages of text under key words, concepts or events. We undertook this process, as referenced in the manuscript, following guidance for qualitative content analysis from a well-established textbook on qualitative data analysis by Margrit Schreier. We also followed Pat Bazeley’s recommendations to pay particular attention to divergent views, negative cases, and pre-existing theoretical literature throughout the analysis.

Section results:

Number of participants belonging to specific groups should be provided to explain the efficacy of the approach, demonstrating the prevalence of specific answers. Moreover, a statistical approach to analyse the groups should be conducted.

Page 9 – lines 377 -380: Was a validated questionnaire proposed to participants to evaluate their responses? Were criteria fixed to discriminate positive answers from negative answers?

Again, we would like to thank you for the above two thought-provoking comments. Indeed, it was also important to us to analyse our pilot intervention using quantitative evaluation techniques. We did this in our first article on this intervention, which has been published (https://www.mdpi.com/1660-4601/16/9/1616). Respectfully, we feel the comments here are more relevant for a quantitative study design than for our current manuscript which analyses text-based data gathered in focus group discussions. Still, we hope to clarify our approaches here for you.

In this study, we consulted a convenience sample of physicians and nurses who had the time to stay behind after the intervention workshops to talk to us. There is only a small number of participants and there is no value in statistically analysing the participant characteristics. It is also not relevant for this qualitative study to use a validated questionnaire, nor to quantify the participants answers (e.g. to analyse positive and negative responses). Rather, we used a semi-structured topic guide (e.g. an outline of topics) in an exploratory manner, meaning the guide allowed the interviewer to probe beyond the participants’ initial answers, seeking clarification and elaboration. This kind of question guide allows for more open discussion and greater depth and personal detail than is the case with quantitative surveys or interviews.

Finally, it is true that some qualitative researchers count the number of participants who speak about a particular topic/theme/category, in their presentation of study results. However, adding frequency counts is still relatively uncommon in qualitative research and there is no consensus about this practice. In this manuscript, we do not feel it is relevant to count the number of participants holding particular beliefs. Such an approach is not in line with our aim to access and describe the subjective perspectives of our research participants. However, we have complemented this manuscript with our (previously published) quantitative evaluation of the pilot intervention.

Section “discussion”:

Page 10 – line 413: Please, provide references. Thank you. We have done this now (see line 472).

Page 10 – line 416: What does “many” mean? As above, it is not relevant to include a particular count of participants and we don’t see any problem in using words like “many” when it is reflective of the data. However, we appreciate the interpretation challenge and have revised the word “many” to “a majority”, which is accurate and hopefully more clarifying (see line 476).

Page 10 – line 433: Please, follow the guideline for authors in providing in-text- citations Amended, thank you (line 493).

Page 10 – lines 433-435: Please, explain further the concept providing scientific evidences supported by references. These are principles at the core of many cognitive-behavioural psychology interventions, largely stemming from Albert Bandura's scholarship. We added a reference to one of Bandura’s key publications on the theory of self-efficacy / the agentive self (see line 444).